# Evidence that human oral glucose detection involves a sweet taste pathway and a glucose transporter pathway

Paul A. S. Breslin[1,2]*, Akiko Izumi[3], Anilet Tharp[2], Tadahiro Ohkuri[3], Yoshiaki Yokoo[3], Linda J. Flammer[2], Nancy E. Rawson[2], Robert F. Margolskee[2]

**1** Department of Nutritional Sciences, Rutgers University, New Brunswick, NJ, United States of America, **2** Monell Chemical Senses Center, Philadelphia, PA, United States of America, **3** Suntory Beverage & Food Limited, Tokyo, Japan

* breslin@monell.org

**Data Availability Statement:** All of the data are included with this submission as an excel file labeled supporting material.

## Abstract

The taste stimulus glucose comprises approximately half of the commercial sugar sweeteners used today, whether in the form of the di-saccharide sucrose (glucose-fructose) or half of high-fructose corn syrup (HFCS). Therefore, oral glucose has been presumed to contribute to the sweet taste of foods when combined with fructose. In light of recent rodent data on the role of oral metabolic glucose signaling, we examined psychopharmacologically whether oral glucose detection may also involve an additional pathway in humans to the traditional sweet taste transduction via the class 1 taste receptors T1R2/T1R3. In a series of experiments, we first compared oral glucose detection thresholds to sucralose thresholds without and with addition of the T1R receptor inhibitor Na-lactisole. Next, we compared oral detection thresholds of glucose to sucralose and to the non-metabolizable glucose analog, α-methyl-D-glucopyranoside (MDG) without and with the addition of the glucose co-transport component sodium (NaCl). Finally, we compared oral detection thresholds for glucose, MDG, fructose, and sucralose without and with the sodium-glucose co-transporter (SGLT) inhibitor phlorizin. In each experiment, psychopharmacological data were consistent with glucose engaging an additional signaling pathway to the sweet taste receptor T1R2/T1R3 pathway. Na-lactisole addition impaired detection of the non-caloric sweetener sucralose much more than it did glucose, consistent with glucose using an additional signaling pathway. The addition of NaCl had a beneficial impact on the detection of glucose and its analog MDG and impaired sucralose detection, consistent with glucose utilizing a sodium-glucose co-transporter. The addition of the SGLT inhibitor phlorizin impaired detection of glucose and MDG more than it did sucralose, and had no effect on fructose, further evidence consistent with glucose utilizing a sodium-glucose co-transporter. Together, these results support the idea that oral detection of glucose engages two signaling pathways: one that is comprised of the T1R2/T1R3 sweet taste receptor and the other that utilizes an SGLT glucose transporter.

**Funding:** This research was funded in part by a grant from the Suntory Global Innovation Center Limited to PASB. The funder consulted on the general conception of the study and provided support in the form of salaries for the authors (PASB, AT, LJF), and a scientist from Suntory (AI) helped collect data under the supervision of the Monell Center. The funder did not play a role in the specific study design, data analyses, decision to publish, or preparation of the manuscript.

**Competing interests:** The authors declare that we have no competing interests in the work described here. The participation of the Suntory Global Innovation Center Limited does not alter our adherence to PLOS ONE policies on sharing data and materials.

## Introduction

The oral detection of sugars is presumed to occur by activation of a class 1 taste receptor heteromer, TAS1R2-TAS1R3, sometimes referred to as a 'sweetener receptor' [1–5]. Notions of how human sweet taste operates are largely based on mouse sweetener perceptual genetics and physiology [6, 7]. By genetic homology, binding to and activation of this receptor is believed to stimulate perceived sweetness in humans as well. In support of this idea, mutations in regulatory regions of the human T1R receptor genes are associated with slightly altered sensitivity to sucrose [8, 9]. However, more recent genetic studies of human sweet taste perception and intake have not replicated these observations [10]. Based on the assumption that the human preference for sugary foods and beverages is driven by stimulation of sweet taste orally, the design and creation of artificially sweetened beverages occurred over 100 years ago by Hyman Kirsch in 1904 who made No-Cal ginger ale with calcium cyclamate to treat diabetics [11, 12]. Despite the century-long refinement of no-calorie or low-caloric sweeteners in beverages, diet sodas have never captured a major share of the beverage market [13]. Reasons for this are presently unknown, but have been attributed to: 1) non-sucrose-like sweetness, 2) non-sucrose-like bitterness and other side tastes, 3) non-sucrose-like temporal profile of sweetness (lingering taste) [14], 4) customer fears of artificial ingredients in foods (naturalism bias) [15], and 5) concerns for increased risk of medical pathologies from use, such as cancer [16, 17]. We hypothesize an additional explanation that sugars may engage a second oral signaling pathway for calories that noncaloric sweeteners fail to engage [18, 19].

Recently, mouse taste bud cells were found to contain many of the same molecular components as do pancreatic beta islet of Langerhans cells, which use a multi-step metabolic signaling pathway to detect glucose in the blood [19]. Beta islet cells indicate increases in blood glucose via: A) the transport of glucose into the cells via molecular carriers such as the sodium-glucose co-transporters (SGLTs) along with other transporters, such as GLUTs, B) the oxidation of glucose to produce several ATP molecules via i) glycolysis involving glucokinase and the production of ATP and pyruvate and ii) oxidative metabolism of the pyruvate via Krebs cycle in mitochondria to yield additional ATP, and lastly, C) the closing of potassium channels that are gated by ATP ($K_{ATP}$ channel) [20]. The presence of glucose transporters [18], glucokinase, and the ATP gated potassium channels $K_{ATP}$ (Kir6.1 and SUR) have all been identified in mouse taste bud cells [19]. Thus, it appears that taste bud cells in the mouth of mice are capable of identifying when a "sweetener" is metabolizable [19, 21]. This system appears to have functionality in mice, as oral stimulation with sugars in the absence of T1R2-T1R3 taste receptors continues to elicit anticipatory insulin responses to sugar [22].

Whether a similar glucose metabolic signaling pathway exists and is functional within human oral taste cells and whether it impacts sugar perception has not been determined. Were such a signaling system to exist and be functional in the human mouth, it would have implications for oral signaling of the metabolizable sugar glucose and could help explain preference for sugared beverages over non-caloric sweetener beverages (cf. [23]). In the present study, we conducted several psychopharmacological experiments to identify whether a second signaling pathway exists in human mouths for sugars such as glucose, but not for non-caloric sweeteners, such as sucralose, and whether this pathway may involve the sodium-glucose-linked co-transporters (SGLTs) as an initial step in this additional signaling pathway. Support for the role of the specific SGLT transporters in glucose taste come from our use of added sodium, which is co-transported with glucose, use of the SGLT inhibitor phlorizin, use of the sugar fructose, which is not transported by SGLT, and use of the glucose analog α-methyl-D-glucopyranoside (MDG) which is co-transported with sodium by SGLT, but is not metabolized to produce ATP [24].

## Materials and methods

### Experiment 1

**Hypothesis.**   We hypothesize that glucose engages a dual oral signal comprised of i) T1R receptor-based signaling and ii) a form of oral metabolic-pathway signaling that involves transport of glucose by an SGLT, whereas sucralose engages only the T1R receptor signaling. Therefore, we predicted that inhibiting the T1R2-T1R3 taste signal with the T1R inhibitor Na-lactisole would have a greater impact (raising the threshold concentration) on sucralose than it would on glucose detection thresholds.

**Ethics statement.**   All research was conducted according to the principles expressed in the Declaration of Helsinki, and approved by an Institutional Review Board at the University of Pennsylvania (IRB #701334). Subjects provided written, informed consent on forms approved by the Institutional Review Board prior to participation.

**Subjects.**   Twelve healthy adults (7 female, 5 male; mean age = 39 years, S.D. = 13) participated. Subjects were recruited from the surrounding community, and paid for their time.

**Stimulus materials.**   Stimuli included filtered water (Milli-Q Water Purification System); serial dilutions of D-(+)-glucose (> 99.5%, Sigma-Aldrich, USA) in filtered water in 1/8 log steps ranging from 0.73–73 mM; serial dilutions of sucralose (> 98.0%, Sigma-Aldrich, China) in filtered water in 1/8 log steps ranging from 2.0–200 μM; serial dilutions of glucose in filtered water in 1/8 log steps ranging from 0.73–412 mM with the addition of 2 mM Na-lactisole (Sodium 2-[4-methoxyphenoxy propionate], Endeavor Specialty Chemicals, UK); serial dilutions of sucralose in filtered water in 1/8 log steps ranging from 2.0–1124.7 μM and 2 mM Na-lactisole; serial dilutions of glucose in filtered water in 1/8 log steps ranging from 0.73–73 mM with the addition of 2mM NaCl (>99.0% sodium chloride, Fisher Scientific, USA); serial dilutions of sucralose in filtered water in 1/8 log steps ranging from 2.0–200 μM and 2 mM NaCl; 0.8 mM Na-lactisole used as a rinse. Stimuli were presented at room temperature.

**Detection threshold method.**   Subjects completed six conditions each with a replicate. The conditions measured absolute detection thresholds for i) glucose and ii) sucralose alone, iii & iv) each with 2 mM Na-lactisole added, and v & vi) each with 2 mM NaCl to control for the sodium associated with lactisole. In these conditions, subjects rinsed with water before and between samples.

Each subject was instructed to refrain from smoking, eating, chewing gum, and drinking anything, except water, for one hour before participation. At the start of each session subjects rinsed their mouths with water 4 times for 30 seconds each and then expectorated for a total rinse time of 2 minutes. They were presented with two 10 ml samples. In the glucose condition the samples were water and glucose. While wearing nose clips, they put the whole sample in their mouth and after 1–2 seconds expectorated and then rinsed with water. Next, they repeated the tasting with the second sample. Their task was to select the sample that is different from water in a two-alternative forced-choice (2-AFC) trial. If unsure, they were instructed to guess. For the glucose + Na-lactisole condition, the samples were water + 2 mM Na-lactisole and glucose + 2 mM Na-lactisole. For the glucose + NaCl condition the samples were water + 2 mM NaCl and glucose + 2 mM NaCl. For these conditions their task was to identify which sample was stronger. Again, if unsure they were instructed to guess. The same scenario was repeated using sucralose in place of glucose comprising three glucose conditions and three sucralose conditions.

Detection thresholds were measured using a modified staircase method. Starting with the average threshold for each test sample (7.33 mM for glucose, 20 μM for sucralose) subjects made their selection (either different from water or the stronger sample depending on the test condition). If their response was correct, they were presented with the same concentration

until they gave 4 correct responses. If their response was incorrect, the next higher concentration was presented in ascending order until 4 correct responses were given in a row at the same concentration. After 4 correct responses, a descending order of concentrations was presented until an incorrect response was given. This pattern was followed until subjects completed 5 reversals in concentration direction (ascending to descending or descending to ascending). If the 5 reversals spread over more than 3 concentration-steps, however, testing continued until the 5 reversals remained within a three concentration-step range in order to clamp variability and avoid random walks on the staircase. The concentrations of the last 4 reversals were averaged to calculate each subject's absolute detection threshold.

**Sweet water taste control.**    Rinsing with Na-lactisole may result in what is known as "sweet water taste" [25]. That is, after Na-lactisole treatment, plain water rinses are sometimes perceived as sweet. To determine if our procedure gave rise to this phenomenon, we measured subject's detection thresholds for both glucose and sucralose under three conditions: 1) when presented against water, 2) with the addition of 2 mM Na-lactisole, and 3) with 0.8 mM Na-lactisole rinses between samples to inhibit sweet water taste. The same twelve subjects (7 female, 5 male; mean age = 39 years, S.D. = 13) that participated in Experiment 1 were tested in these three conditions in duplicate. The concentration of 0.8 mM Na-lactisole for the rinse was selected in preliminary studies to prevent sweet water taste from 2 mM Na-lactisole treatment. See S1 Fig.

A repeated measures analysis of variance for the glucose detection thresholds revealed a significant effect of condition $F_{(2, 66)} = 31.50$, $p < .00001$. Post-hoc Tukey HSD analyses revealed the detection thresholds for all three conditions (water, Na-lactisole, and Na-lactisole rinse) were significantly different from one another, $p < .001$, with the detection threshold for glucose without treatment as the lowest and the detection threshold with the 0.8 mM Na-lactisole rinses between stimuli as the highest. Therefore, there was no evidence for an effect of sweet water taste following Na-lactisole treatment in this procedure. There was no effect of replication, nor an interaction between condition and replication. The repeated measures analysis of variance for the sucralose detection thresholds also revealed a significant effect of condition $F_{(2, 66)} = 21.42$, $p < .00001$. Post-hoc Tukey HSD analyses revealed the detection thresholds for Na-lactisole treatment and the Na-lactisole treatment with rinse were significantly higher than the detection threshold for sucralose alone, $p < .001$. Again, there was no evidence for a sweet water taste following Na-lactisole treatment in this procedure. There was no effect of replication, nor an interaction between condition and replication.

## Experiment 2A & B

**Hypothesis.**    Based upon the outcomes of Experiment 1, we hypothesized that if the additional signaling pathway for glucose involves transport into cells via the sodium glucose cotransporter (SGLT), then adding NaCl to both glucose and MDG should enhance detection (lower detection threshold), but not enhance sucralose detection. MDG is transported by SGLT but is not metabolizable into ATP. Thus, MDG will determine whether movement of sodium with MDG by an SGLT is sufficient to enhance detection.

**Subjects.**    The same subjects who participated in Experiment 1, participated in the detection threshold experiment comparing glucose to sucralose (Experiment 2A). A different group of subjects participated in the experiment comparing detection thresholds of glucose and MDG (Experiment 2B). Eleven healthy adults (7 female, 4 male; mean age = 44 years, S.D. = 12) participated. Subjects were recruited from among the surrounding community, and paid for their time.

**Stimulus materials.**   Stimuli included filtered water (Milli-Q Water Purification System); serial dilutions of D-(+)-glucose (> 99.5%, Sigma-Aldrich, USA) in filtered water in 1/8 log steps ranging from 0.73–73 mM; serial dilutions of D-(+)-glucose in filtered water in 1/8 log steps ranging from 0.73–73 mM with 20 mM NaCl (>99.0% sodium chloride, Fisher Scientific, USA) added; serial dilutions of sucralose (> 98.0%, Sigma-Aldrich, China) in filtered water in 1/8 log steps ranging from 2.0–200 μM; serial dilutions of sucralose in filtered water in 1/8 log steps ranging from 2.0–200 μM with 20 mM NaCl added; serial dilutions of α-methyl-D-gluco-pyranoside (MDG) (> 99%, Sigma-Aldrich, China) in filtered water in 1/8 log steps ranging from 0.73–73 mM; serial dilutions of MDG in filtered water in 1/8 log steps ranging from 0.73–73 mM with 20 mM NaCl added. Stimuli were presented at room temperature.

**Detection threshold method.**   For Experiment 2A, subjects completed 4 conditions each with a replicate. The conditions measured detection thresholds for glucose and sucralose each with and without the addition of 20 mM NaCl. The same threshold measurement protocol described in Experiment 1 was followed for Experiment 2A. Detection thresholds were measured using a modified staircase method with 5 reversals. In the glucose condition, the sample pairs were deionized filtered water versus glucose or deionized filtered water with 20 mM NaCl added versus glucose with 20 mM NaCl added. In the sucralose condition, the sample pairs were deionized filtered water versus sucralose or deionized filtered water with 20 mM NaCl added versus sucralose with 20 mM NaCl added. The subject's task was to identify which sample in the pair was stronger. If unsure, they were instructed to pick one.

For Experiment 2B, subjects also completed 4 conditions each with a replicate. The conditions measured detection thresholds for glucose and MDG each with and without the addition of 20 mM NaCl. The same threshold measurement protocol described in Experiment 1 was followed for Experiment 2B. Detection thresholds were measured using a modified staircase method with 5 reversals. In the glucose condition, the sample pairs were deionized filtered water versus glucose or deionized filter water with 20 mM NaCl added versus glucose with 20 mM NaCl added. In the MDG condition the samples were deionized filtered water versus MDG or deionized filtered water with 20 mM NaCl added versus MDG with 20 mM NaCl added. The subject's task was to identify which sample in the pair was stronger. If unsure, they were instructed to pick one.

## Experiment 3

**Hypothesis.**   Based upon the outcome of Experiments 2A and 2B, we hypothesized if oral glucose signaling involves an SGLT, then the pharmacological SGLT inhibitor, phlorizin, should increase absolute detection thresholds for glucose and MDG, but not for sucralose, and have no effect on fructose that is transported by GLUT5 instead of SGLT [26].

**Subjects.**   Eleven of the same subjects that participated in Experiment 1participated in the glucose and sucralose segments of Experiment 3 (6 female, 5 male; mean age = 39 years, S.D. = 13). For the MDG segment subjects were the same individuals that participated in Experiment 2B (7 females and 4 males with a mean age = 44 years with S.D. = 12). Finally, for the fructose segment subjects included 7 females and 5 males with a mean age = 36 years with S.D. = 12.

**Stimulus materials.**   Stimuli included filtered water (Milli-Q Water Purification System); serial dilutions of D-(+)-glucose (> 99.5%, Sigma-Aldrich, USA) in filtered water in 1/8 log steps ranging from 0.73–73 mM; serial dilutions of D-(+)-glucose in filtered water in 1/8 log steps ranging from 0.73–73 mM with 20 mM NaCl (>99.0% sodium chloride, Fisher Scientific, USA) added; serial dilutions of D-(+)-glucose in filtered water in 1/8 log steps ranging from 0.73–73 mM with 20 mM NaCl and 0.2 mM phlorizin (>98.0%, Cayman Chemical Company,

USA) added; serial dilutions of sucralose (> 98.0%, Sigma-Aldrich, China) in filtered water in 1/8 log steps ranging from 2.0–200 μM; serial dilutions of sucralose in filtered water in 1/8 log steps ranging from 2.0–200 μM with 20 mM NaCl added; serial dilutions of sucralose in filtered water in 1/8 log steps ranging from 2.0–200 μM with 20 mM NaCl and 0.2 mM phlorizin added; serial dilutions of α-methyl-D-glucopyranoside (MDG) (> 99%, Sigma-Aldrich, China) in filtered water in 1/8 log steps ranging from 0.73–73 mM; serial dilutions of α-methyl-D-glucopyranoside (MDG) in filtered water in 1/8 log steps ranging from 0.73–73 mM with 20 mM NaCl added; serial dilutions of α-methyl-D-glucopyranoside (MDG) in filtered water in 1/8 log steps ranging from 0.73–73 mM with 20 mM NaCl and 0.2 mM phlorizin added; serial dilutions of D-(-)-fructose (> 99%, Sigma-Aldrich, USA) in filtered water in 1/8 log steps ranging from 0.56–56 mM; serial dilutions of D-(-)-fructose in filtered water in 1/8 log steps ranging from 0.56–56 mM with 20 mM NaCl added; serial dilutions of D-(-)-fructose in filtered water in 1/8 log steps ranging from 0.56–56 mM with 20 mM NaCl and 0.2 mM phlorizin added. Stimuli were presented at room temperature.

**Detection threshold method.** There were four segments in Experiment 3. One for each sweetener: glucose, fructose, MDG, and sucralose. In each segment, subjects completed 3 conditions each with a replicate. The conditions measured detection thresholds for the sweetener alone, with the addition of 20 mM NaCl, and with the addition of 20 mM NaCl and 0.2 mM phlorizin. The same protocol described in Experiments 1 and 2 was followed for Experiment 3. Detection thresholds were measured using a modified staircase method with 5 reversals. For each sweetener type (glucose, fructose, MDG, and sucralose the sample pairs were filtered water versus sweetener alone, filtered water with 20 mM NaCl versus sweetener with 20 mM NaCl, filtered water with 20 mM NaCl + 0.2 mM phlorizin added versus sweetener with 20 mM NaCl + 0.2 mM phlorizin added. The subject's task was to identify which sample was stronger. If unsure, they were instructed to pick one.

## Results

All analyses were conducted using Statistica software (Version 13.5.0.17, Tibco), using an alpha value of <0.05 for allowing Type I errors.

### Experiment 1

Factorial repeated-measures analysis of variance (ANOVA) were conducted to compare the main effects of condition (water, Na-lactisole, NaCl), and replication (1, 2) and the interaction effect on detection thresholds for both glucose and sucralose. There was a main effect of condition for each sweetener such that the Na-lactisole treatment significantly raised threshold levels. For glucose $F_{(2, 66)} = 24.86$, $p < .0001$, post-hoc Tukey HSD analyses showed the detection threshold with Na-lactisole (mean = 41.8 mM, S.D. = 25.9) was significantly higher than both glucose alone (mean = 13.3 mM, S.D. = 5.5) and glucose with NaCl (mean = 13.8 mM, S.D. = 4.6), $p < .001$. For sucralose $F_{(2, 66)} = 33.00$, $p < .0001$, post-hoc Tukey HSD analyses showed the detection threshold with Na-lactisole (mean = 100.0 μM, S.D. = 73.7) was significantly higher than both sucralose alone (mean = 11.9 μM, S.D. = 5.4) and sucralose with NaCl (mean = 11.9 μM, S.D. = 5.0), $p < .001$. The water and NaCl conditions did not differ from each other for either sweetener. The 2 mM NaCl was added to control for the effects of 2 mM sodium from the Na-lactisole. There was no effect of replication nor an interaction between condition and replication for either sweetener. As expected, the Na-lactisole treatment decreased sensitivity to the sweeteners glucose and sucralose (See Fig 1A).

Also as expected the Na-lactisole treatment had a significantly larger impact on sucralose than on glucose. After Na-lactisole treatment and with water rinsing between samples, the

A

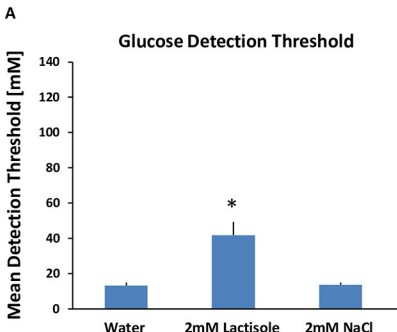

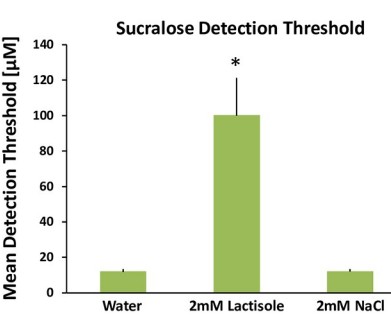

B

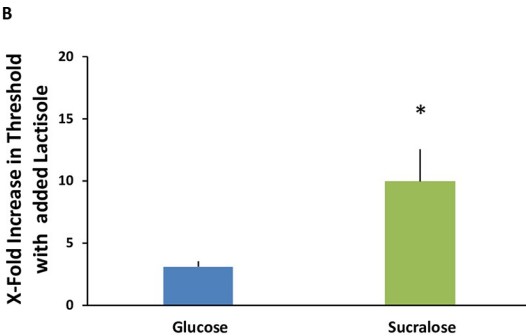

**Fig 1. A. Detection thresholds for glucose [mM] and sucralose [μM] with and without Na-lactisole.** There are two bar charts in Fig 1A. The chart on the left (blue) shows detection thresholds for glucose. The chart on the right (green) shows detection thresholds for sucralose. Each chart contains three bars. From left to right the bars are: 1) Threshold with no treatment, 2) Threshold with 2mM Na-lactisole, 3) Threshold with 2mM NaCl (to control for sodium present in lactisole). All treatments were paired with water rinses between stimuli. Twelve subjects participated in all conditions in duplicate. The error bars are standard errors of the mean. * indicates a significant difference $p < .001$. **B. Impact of Na-lactisole on glucose and sucralose detection thresholds.** The "fold increase" in threshold concentration is shown by taking the threshold concentration when treated with Na-lactisole and dividing it by the threshold concentration when untreated. The bar on the left (blue) represents glucose and the bar on the right (green) represents sucralose. The treatment concentration of Na-lactisole was 2 mM and the untreated condition contained the same level of sodium via the addition of 2 mM NaCl. All treatments were paired with water rinses between stimuli. Twelve subjects participated in all conditions in duplicate. The error bars are standard error of the mean. * indicates a significant difference, $p < .05$.

'fold' increase in threshold concentration for glucose was x 3.1 and for sucralose x 10.0, t (11) = -2.99, p = 0.012 (See Fig 1B). Overall, our hypothesis is supported by these data; Na-lactisole interfered with detecting sucralose approximately three times more than it interfered with glucose detection.

## Experiment 2A

Factorial repeated-measures ANOVA were conducted to compare the main effects of condition (with and without NaCl), and replication (1, 2) and the interaction effect on detection thresholds for both glucose and sucralose. For each sweetener there was a significant effect of condition. For glucose the addition of NaCl significantly lowered the detection threshold (glucose mean = 13.3 mM, S.D. = 5.5 while glucose + NaCl mean = 7.2 mM, S.D. = 1.6), F (1, 44) = 22.71, p < .0001. Conversely, for sucralose the addition of NaCl significantly raised the detection threshold (sucralose mean = 11.9 μM, S.D. = 5.4 while sucralose + NaCl mean = 19.1 μM, S.D. = 7.7), F (1, 44) = 13.88, p < .001. Thus, the addition of 20 mM NaCl enhanced sensitivity to glucose (lowered threshold concentration) by 46% on average, whereas the addition of 20 mM NaCl diminished sensitivity to sucralose (raised threshold concentration) by 161% on average. There was no effect of replication nor an interaction between condition and replication for either sweetener (See Fig 2).

The hypothesis that NaCl would enhance sensitivity to glucose but not sucralose is supported by these data. These data are consistent with SGLT participating in glucose signaling by moving glucose and sodium together into the cells.

## Experiment 2B

A factorial repeated-measures ANOVA was conducted to compare the main effects of sweetener (glucose or MDG), condition (with and without NaCl), and replication (1, 2) and the interaction effects on the detection thresholds. The ANOVA revealed there was not a significant difference between the sweeteners, thresholds are similar, but there was a main effect of condition F (3, 80) = 6.28, p < .001. Post-hoc Tukey HSD analyses revealed the addition of NaCl lowered the detection thresholds for each sweetener, glucose mean = 12.6 mM, S.D. =

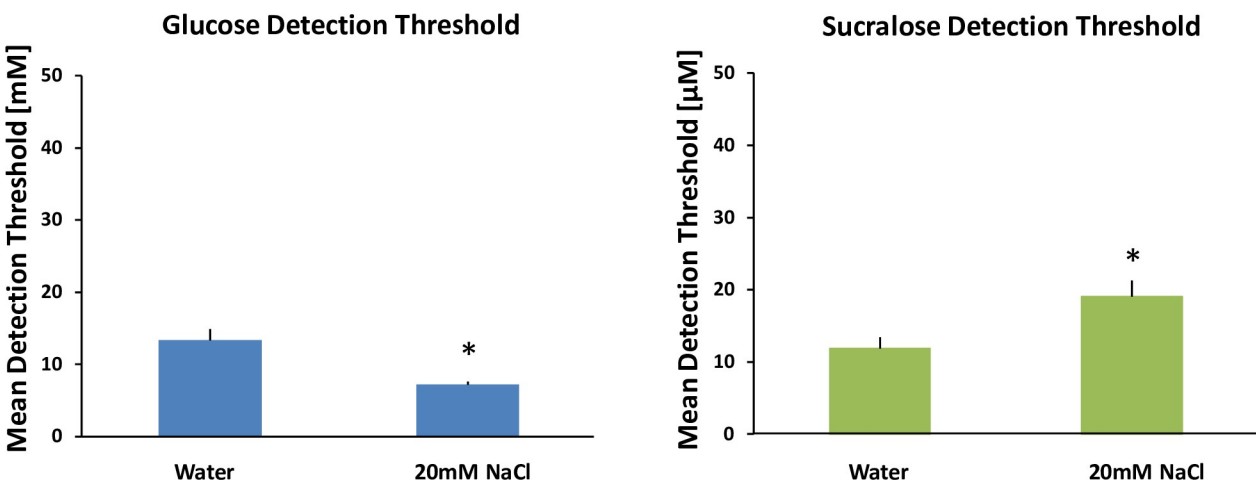

**Fig 2. Mean glucose [mM] and sucralose [μM] detection thresholds with added NaCl.** There are two bar charts in Fig 2. The left chart shows detection thresholds for glucose (blue). The right chart shows detection thresholds for sucralose (green). Each chart contains two bars. From left to right the bars are: 1) Detection threshold, 2) Detection threshold with 20mM NaCl added. 20mM NaCl was selected to match the concentration of glucose near threshold ~ 20mM. Thus, for each glucose molecule there would be a Na+ ion to be co-transported in theory. All treatments were paired with water rinses between stimuli. Twelve subjects participated in all conditions in duplicate. The error bars are standard error of the mean. * indicates significant difference, $p < .0001$.

4.2 while glucose + NaCl mean = 9.3 mM, S.D. = 2.8, $p < .01$; MDG mean = 11.7 mM, S.D. = 2.9 while MDG + NaCl mean = 9.0 mM, S.D. = 1.8, $p < .05$. There was no effect of replication nor any significant interaction effects (See Fig 3). The lowering of the detection thresholds for both glucose and MDG by the addition of NaCl is further evidence that SGLT is involved in moving glucose and its analog into the cell, suggesting a second signaling pathway. Furthermore, it suggests that movement of MDG with sodium by SGLT is sufficient to alter thresholds and ATP generation is not required.

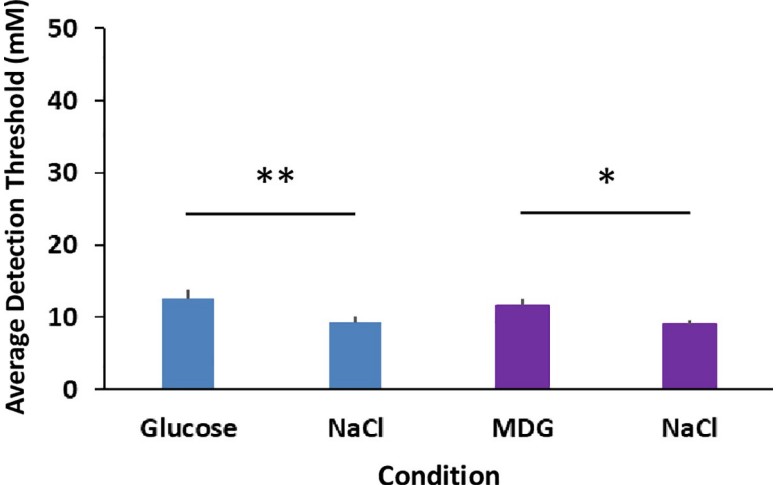

**Fig 3. Comparison of detection thresholds for glucose and MDG with and without NaCl.** There are four bars in the chart. The first bar starting from the left shows the average detection threshold for glucose. The second bar shows the average detection threshold for glucose with the addition of 20 mM NaCl. The third bar shows the average detection threshold for MDG. The fourth bar on the right shows the average detection threshold for MDG with 20 mM NaCl. All treatments were paired with water rinses between stimuli. Eleven subjects participated in all conditions in duplicate. The error bars are standard errors of the mean. * indicates a significant difference $p < .05$; ** indicates a significant difference $p < .01$.

## Experiment 3

Factorial repeated-measures ANOVA were conducted to compare the main effects of condition (water, NaCl, and phlorizin), and replication (1, 2) and the interaction effects on detection thresholds for glucose, MDG, fructose, and sucralose. The results for glucose revealed a significant main effect for condition $F_{(2, 60)} = 58.27$, $p < .00001$. Post-hoc Tukey HSD analyses showed the detection threshold for glucose with NaCl and phlorizin added (mean = 39.9 mM, S.D. = 16.8) was significantly higher than the detection threshold for glucose alone (mean = 13.5 mM, S.D. = 5.7) and for glucose with NaCl (mean = 7.2 mM, S.D. = 1.6 mM), $p < .001$. Similar to glucose, there was a main effect for condition for MDG, $F_{(2, 60)} = 62.05$, $p < .00001$. Post-hoc Tukey HSD analyses showed the detection threshold for MDG with NaCl and phlorizin added (mean = 37.6 mM, S.D. = 15.4) was significantly higher than the detection threshold for MDG alone (mean = 11.7 mM, S.D. = 2.9) and for MDG with NaCl (mean = 9.0 mM, S.D. = 1.8), $p < .001$. For fructose, there were no significant differences in condition. Finally, for sucralose there was a main effect for condition $F_{(2, 60)} = 3.45$, $p < .05$. Post-hoc Tukey HSD analyses showed the detection threshold for sucralose with NaCl (mean = 18.3 mM, S.D. = 7.5 mM) was significantly higher than sucralose alone, (mean = 12.2 mM, S.D. = 5.6 mM) $p < .05$, but no significant differences for the phlorizin condition (mean = 16.2 mM, S.D. = 9.6 mM). There was no effect of replication nor an interaction between condition and replication for any of the sweeteners (See Fig 4).

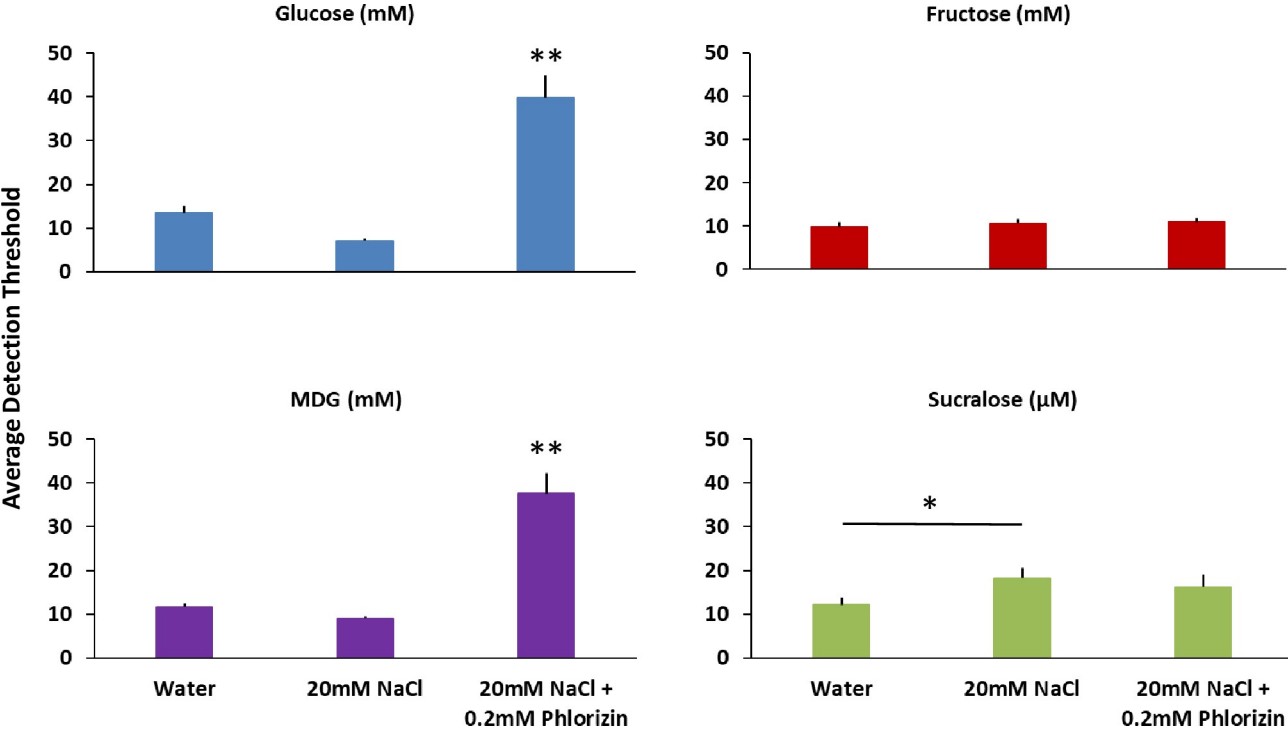

**Fig 4. Impact of phlorizin on glucose, fructose, MDG and sucralose detection thresholds.** There are four bar charts in Fig 4. The chart with the blue bars show average detection thresholds for glucose in mM. The chart with the red bars show average detection thresholds for fructose in mM. The chart with the purple bars show average detection thresholds for MDG in mM. The chart with the green bars show average detection thresholds for sucralose in μM. Each chart contains three bars. From left to right the bars show: 1) Detection threshold, 2) Detection threshold with 20mM NaCl added, 3) Detection threshold with 20mM NaCl and 0.2mM phlorizin added. All treatments were paired with water rinses between stimuli. Twelve subjects participated in the fructose and sucralose testing and eleven subjects participated in the glucose and MDG testing. All subjects completed two replications. Error bars are standard errors of the mean. * indicates a significant difference, $p < .05$, ** indicates a significant difference $p < .001$.

Phlorizin increased the absolute detection thresholds for both glucose and MDG in 100% of the subjects (11 of 11) by approximately a 3-fold increase on average. Whereas, phlorizin had no effect on absolute detection thresholds for fructose or sucralose. The pharmacological SGLT inhibitor phlorizin had a profound impact on impeding every subject's ability to detect glucose and MDG but did not impede their detection of fructose nor sucralose, further supporting the idea that oral glucose signaling involves a second pathway involving SGLT.

## Discussion

These rinse-and-expectorate experiments point to the ability of humans to sense glucose orally via a signaling pathway that includes the sodium-glucose cotransporters (SGLTs). The SGLT is an initial step in what is described as the glucose metabolic signaling pathway, and our data suggest this exists in parallel to the traditional T1R sweet taste pathway for oral glucose sensing in humans. In the first experiment, we used the T1R (class 1 taste receptor) inhibitor Na-lactisole [25, 27, 28] to block sweet taste. Na-lactisole had a much larger impact on sucralose detection thresholds than it did on glucose thresholds Fig 1A. Fig 1B illustrates that detection thresholds for sucralose increased in concentration (indicating decreased sensitivity) by more than 8-fold in the presence of Na-lactisole, whereas glucose thresholds only increased by 3-fold. We interpret this as supporting the idea that there are two sugar sensing coding channels in the mouth: one for signaling the sweet taste of sugars and a second that we hypothesized is based on the metabolic signaling pathway for saccharides. Sucralose only engages the T1R-sweet taste receptor, so inhibition of this receptor has a larger impact on its detection. Glucose, however, engages both signaling channels and in the presence of Na-lactisole continues to be signaled by a second pathway.

In a second experiment, we tested whether the SGLT glucose transporters are involved with the initial step of moving the saccharide into the cell. Since the SGLTs must move sodium together with glucose, we hypothesized that adding sodium (NaCl) at levels comparable to the detection thresholds for glucose (~20 mM at the high end) would lower the threshold for glucose but not for the non-caloric sweetener sucralose by enhancing glucose transport into the cells. The SGLT1 transporter moves two sodium ions for each glucose molecule and the SGLT2 transporter moves one sodium ion for each glucose molecule [29, 30]; therefore, this concentration of sodium would improve movement for both transporters. We found that glucose thresholds decreased (indicating increased sensitivity) in the presence of 20 mM NaCl by approximately 50%, but sucralose thresholds did not decrease with added NaCl (See Fig 2). In fact, the addition of NaCl interfered with sucralose sensitivity and raised detection threshold concentrations modestly. This is most likely due to the weak salty taste of 20 mM NaCl cognitively suppressing detection of sucralose by acting as a sweet-taste masking agent. This influence of the salty taste would also have occurred for glucose, but the impact of added Na on the SGLT mechanism overcame this masking effect.

In the third experiment, we used phlorizin as an SGLT inhibitor [31] to interfere with glucose signaling in the mouth. Whereas the addition of sodium was expected to enhance glucose signaling and decrease detection thresholds of glucose, the addition of phlorizin was predicted to interfere with glucose signaling and increase glucose detection thresholds. We found the addition of 2 mM phlorizin increased glucose detection thresholds compared to the glucose + NaCl condition by approximately 5-fold, whereas sucralose detection thresholds were unaffected by the addition of phlorizin (See Fig 3). Furthermore, there was no effect on fructose thresholds of the addition of phlorizin. This is consistent with glucose taste signaling utilizing an SGLT transporter, but fructose does not use SGLT [26]. Phlorizin is a natural phenolic glucoside chalcone common in apple trees, which when dimerized and oxidized is responsible for

the yellow-brown color of apple juice and cider. It is a non-selective inhibitor of the SGLTs [31], so we cannot conclude from the present data whether one particular SGLT (SGLT1 or SGLT2) is more important for glucose signaling. Note that there are currently six known forms of SGLT transporters, although SGLT3 in humans is thought to be a glucose receptor, not a transporter [32]. There is recent evidence that phlorizin may also inhibit the GLUTs in addition to SGLTs [33]. We do not believe this is reflected in our results, however, as phlorizin had no impact on fructose, supporting the involvement of SGLTs but not GLUT2, GLUT5, or GLUT8. We also saw NaCl enhance glucose thresholds (See Fig 2), further indication of a role for SGLT and not GLUTs in this effect. Regardless, we have established that we can manipulate human oral sensitivity to glucose positively with the addition of sodium and negatively with the addition of phlorizin, all without similar impact on the sweetener sucralose. This bidirectional psychopharmacological approach has strongly implicated the SGLTs as participating in the first step of sugar transport into the cell. Logically, it is likely that other sugars besides glucose, such as galactose, can be transported by SGLT in taste tissue [34], and sugars that do not engage SGLT, such as fructose, may be transported via GLUT2 and GLUT8 in taste tissue [21] and oxidized in taste tissue to produce ATP and close $K_{ATP}$ channels [35].

The glucose metabolic signaling pathway combined with the sweet taste pathway creates a striking parallel between the glucose signaling abilities of the pancreatic beta islet cells of Langerhans and the signaling abilities of the oral cavity (most likely taste tissue) [20, 36]. It has previously been reported that mice possess within oral taste buds: 1) sugar transporters, 2) kinases required to convert sugars into ATP, and 3) an ATP sensor in the form of an ion channel [18, 19]. Whether any of these exist in humans has been previously unknown. In the present work, we also included MDG as a taste stimulus. This glucose analog can be transported by SGLT but does not get oxidized to produce ATP, as it cannot be metabolized [24]. This allows us to distinguish whether the first step in the metabolic signaling pathway can produce oral signals independently of the remainder of the traditional metabolic signaling pathway, such as ATP generation and the closure of $K_{ATP}$ channels. We found that MDG oral detection thresholds were enhanced by the addition of NaCl and impaired by the addition of phlorizin (See Fig 3). This indicates that the transport of sodium by SGLT with glucose and MDG can activate cells irrespective of whether ATP acts on $K_{ATP}$ channels. Logically, the closure of $K_{ATP}$ channels in this signaling pathway may also further activate cells.

Collectively, these data allow us to screen enhancers of glucose signaling via stimulation of the SGLT pathway for their impact on oral sugar signaling. These may involve any form of pharmacological SGLT enhancement including the addition of sodium to glucose, SGLT pharmacological modulation (allosteric or otherwise) to better transport glucose across the membrane, and any glucose mimetic that can be transported into cells via SGLT. We speculate that it may be possible to reduce sugar levels in beverages and foods by enhancing glucose via the metabolic signaling pathway, especially at the level of SGLT transport. We note that SGLT signals may contribute to sweet taste, but may also contribute to an independent non-sweet signal that conveys the presence of glucose [18]. This idea is suggested by work in T1R-Knock-Out mice which showed anticipatory insulin release to a glucose load, but did not show a behavioral preference for glucose in water [22]. It is possible that oral SGLT contributes to glucose reward in humans either directly or indirectly via anticipatory metabolic regulatory reflexes, but note that oral SGLT in mice does not appear to contribute to preference [23]

In summary, the experiments presented here utilizing the T1R inhibitor Na-lactisole, the co-transport agent for SGLT added NaCl, and the SGLT inhibitor phlorizin support the idea that SGLTs are involved in the oral perception of glucose, but not the perception of sucralose or fructose. These studies show that a non-T1R oral signaling pathway profoundly affects

absolute detection of glucose but not sucralose or fructose. Future studies will determine the utility of this SGLT-linked oral glucose signal in human psychology and physiology.

## Supporting information

**S1 Fig. No Evidence of "Sweet Water Taste" following lactisole treatment.** There are two charts. The chart on the left (blue bars) shows average detection thresholds for glucose (mM). The chart on the right (green bars) shows average detection thresholds for sucralose (μM). In each chart there are three bars. From left to right the first bar is the average detection threshold for the sweetener when presented against water. The second bar is the average detection threshold for the sweetener with the addition of 2mM lactisole. The third bar is the average detection threshold for the sweetener with 0.8mM lactisole rinses between samples. The lactisole rinses were to determine if the lactisole treatment gave rise to a "sweet water taste." Twelve subjects were tested in all conditions in duplicate. The error bars are standard errors of the mean. * indicates a significant difference p < .001.
(TIF)

**S1 Data.**
(XLSX)

## Author Contributions

**Conceptualization:** Paul A. S. Breslin, Tadahiro Ohkuri, Yoshiaki Yokoo, Nancy E. Rawson, Robert F. Margolskee.

**Data curation:** Paul A. S. Breslin.

**Formal analysis:** Paul A. S. Breslin, Linda J. Flammer.

**Investigation:** Akiko Izumi, Anilet Tharp.

**Methodology:** Paul A. S. Breslin, Anilet Tharp.

**Project administration:** Paul A. S. Breslin.

**Resources:** Paul A. S. Breslin.

**Supervision:** Paul A. S. Breslin.

**Validation:** Paul A. S. Breslin, Anilet Tharp, Linda J. Flammer.

**Visualization:** Paul A. S. Breslin, Linda J. Flammer.

**Writing – original draft:** Paul A. S. Breslin, Linda J. Flammer.

**Writing – review & editing:** Paul A. S. Breslin, Akiko Izumi, Anilet Tharp, Tadahiro Ohkuri, Yoshiaki Yokoo, Linda J. Flammer, Nancy E. Rawson, Robert F. Margolskee.

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
