## [Decision Letter · Decision Letter 0]

8 Jun 2021

PONE-D-21-16275

Evidence that Human Oral Glucose Detection Involves A Sweet Taste Pathway and A Glucose Transporter Pathway

PLOS ONE

Dear Dr. Breslin,

Thank you for submitting your manuscript to PLOS ONE. After careful consideration, we feel that it has merit but does not fully meet PLOS ONE’s publication criteria as it currently stands. Therefore, we invite you to submit a revised version of the manuscript that addresses the points raised during the review process.

Dear Dr. Breslin and coauthors,

You have done a superb job addressing most of the concerns raised by the reviewers. In particular, I commend you for conducting the additional experiments. As you note, they strengthen your study and provide robust support for the central claim of your study--that SGLT1 is involved in oral glucose detection in humans. This is clearly a novel and important finding. 

Overall, I believe that you have addressed the majority of the reviewers' original concerns. You will see, however, that Reviewer 1 has some remaining concerns that need to be addressed in your revision. I also have a few editorial suggestions. 

1. Your current title is: "Evidence that Human Oral Glucose Detection Involves A Sweet Taste Pathway and A Glucose Transporter Pathway." This title implies that you have identified two pathways for detecting glucose in the oral cavity. This is not the case. What you have established is that one can systematically alter glucose detection thresholds by modulating the activity of SGLT1. While I do not mean to diminish the importance of this finding, you currently do not know whether modulating the activity of SGLT1 alters downstream voltage-sensitive events in the T1r2-r3 pathway or whether it activates a T1r2-r3-independent pathway. Thus, I recommend that you scale back the scope of your title.

2. End of first paragraph of Introduction: You state: "Clearly sugars engage a second signaling pathway for calories that noncaloric sweeteners fail to engage post-prandially." I do not think that this statement follows logically from the five reasons you provide for why "...diet sodas have never captured a major share of the beverage market." Reviewer 1 has raised additional concerns about this statement.

3. Line 1, second paragraph of Introduction: Change "mice" to "mouse"

4. Line 4, second paragraph of Introduction: The following phrase seems awkward: "...the transportation of glucose". I think that it is more conventional to say "...the transport of glucose"

5. Line 5, second paragraph of Introduction: Why don't you refer directly to the GLUTs? That would seem to be clearer than a vague reference to "...other transporters". 

6. Lines 5-6, second paragraph of Introduction: I think that the following wording is accurate: "...the conversion of glucose into several ATP molecules..." Glucose is not actually converted into ATP molecules. The energy in the glucose molecule is used by glycolysis, krebs cycle and oxidative phosphorylation to add high-energy phosphate bonds to AMP and ADP.

7. Line 10, second paragraph of Introduction: I recommend changing "mouse tissue" to "mouse taste bud cells."

8. Lines 10-13, second paragraph of Introduction: I do not think that the prior literature has established that "taste bud cells in the mouth of mice are capable of identifying when a “sweetener” is metabolizable." Instead, I think this literature has established that murine taste bud cells may be able to detect elevations in salivary glucose via a T1r2+r3-independent mechanism.

9. In the last paragraph of the Introduction, you state: "Were such a signaling system to exist and be functional in the human mouth, it would have implications for oral signaling of metabolizable sugars and could help explain why humans overwhelmingly prefer sugared beverages to non-caloric beverages." I recommend that you dial-back this statement a bit for two reasons. First, because your paper addresses threshold (not suprathreshold) effects of SGLT1, you are creating false expectations in your readers. Second, the use of "metabolizable sugar" is a bit confusing. As noted by Reviewer 1, fructose may not be metabolized in taste cells, but it is certainly metabolized in the liver. Thus, fructose is a metabolizable sugar that does not activate SGLT1.

10. I was surprised that you did not incorporate the findings from a recent mouse paper [Yasumatsu K, Ohkuri T, Yoshida R, Iwata S, Margolskee RF, Ninomiya Y (2020) Sodium-glucose cotransporter 1 as a sugar taste sensor in mouse tongue. Acta Physiologica 230: e13560. doi: 10.1111/apha.13529], as it is highly relevant and supportive of your general hypothesis.

We look forward to receiving your revised manuscript.

Kind regards,

John I. Glendinning, PhD

Academic Editor

PLOS ONE

Journal Requirements:

"This research was funded in part by a grant from the Suntory Global Innovation Center Limited

to PASB. The funder consulted on the general conception of the study and provided support in

the form of salaries for the authors (PASB, AT, LJF) and a scientist from Suntory (AI) helped

collect data under the supervision of the Monell Center. The specific roles of the authors are

articulated in the “Author Contributions” section. The funder did not play a role in the specific

study design, data analyses, decision to publish, or preparation of the manuscript."

"The funder did not play a role in the specific study design, data analyses, decision to publish, or preparation of the manuscript.  "

Additionally, because some of your funding information pertains to commercial funding, we ask you to provide an updated Competing Interests statement, declaring all sources of commercial funding.

In your Competing Interests statement, please confirm that your commercial funding does not alter your adherence to PLOS ONE Editorial policies and criteria by including the following statement: "This does not alter our adherence to PLOS ONE policies on sharing data and materials.” as detailed online in our guide for authors  http://journals.plos.org/plosone/s/competing-interests.  If this statement is not true and your adherence to PLOS policies on sharing data and materials is altered, please explain how.

Please include the updated Competing Interests Statement and Funding Statement in your cover letter. We will change the online submission form on your behalf.

Reviewers' comments:

Reviewer's Responses to Questions

**Comments to the Author**

1. Is the manuscript technically sound, and do the data support the conclusions?

Reviewer #1: Yes

2. Has the statistical analysis been performed appropriately and rigorously? 

Reviewer #1: Yes

3. Have the authors made all data underlying the findings in their manuscript fully available?

Reviewer #1: Yes

4. Is the manuscript presented in an intelligible fashion and written in standard English?

Reviewer #1: Yes

5. Review Comments to the Author

Reviewer #1: This revised manuscript is substantially enhanced by the addition of MDG and fructose experiments which support the involvement of SGLT1 sensing in oral glucose detection in humans. However, the characterization of this oral SGLT1-glucose sensing pathway as a "caloric" or " metabolic signaling pathway" is misleading. The SGLT1 pathway does not respond to fructose which is a "caloric" and "metabolizable" sugar. It is also unclear how much sucrose activates the SGLT1 taste pathway given that sucrose must be first hydrolyzed to glucose in the mouth to have this action. The authors should revise their description of the SGLT1 taste pathway. Also, for completeness, the authors should mention that galactose is a ligand for SGLT1 although, given the relatively low amount of free galactose in foods, this sugar may have minimal effects on the oral SGLT1 pathway.

The authors describe MDG as a ligand for SGLT1 but don't mention that it is also a ligand for the T1R2/T1R3 receptor and thus has a sweet taste to humans as it apparently has to mice as judged by their avidity for this glucose analog. The authors speculate that the SGLT1 taste pathway may contribute to the sweetness (or palatability) of sugars in humans and contribute to their preference for sugars over non-nutritive sweeteners. However, this does not appear to be the case in mice. SGLT1 KO and WT mice showed identical preferences for MDG over water in 3-min, 2-bottle tests (Sclafani, Koepsell and Ackroff, 2016). Furthermore, SGLT1 KO and WT mice showed similar preferences for a glucose + saccharin solution over a saccharin in a 1-h choice test. The KO mice consumed less glucose and MDG than WT mice which was attributed to their failure to normally absorb these sugars in the intestine. The authors should acknowledge that the oral glucose-sensing SGLT1 pathway does not appear to mediate oral sugar preference in mice although, as discussed next, it is essential for post-oral glucose preference conditioning. This does not preclude them from speculating about a role of SGLT1 taste pathway in sugar preference in humans.

The following statement in the Introduction should be supported with citations: "Clearly sugars engage a second signaling pathway for calories that noncaloric sweeteners fail to engage post-prandially." Presumably, the authors are referring to rodent and human studies showing that the post-oral actions of glucose and glucose-containing carbohydrates (sucrose, maltodextrin) condition preferences. However, describing this post-prandial pathway as "for calories" is questionable given the many rodent studies showing that glucose is much more effective than fructose in activating the post-prandial pathway. Of particular relevance here are studies showing that mice learn to prefer glucose but not isocaloric fructose over non-nutritive sweeteners (sucralose, Ace K), an outcome thought to be mediated by intestinal SGLT1 signaling (e.g., Sclafani & Ackroff, 2017; Tan et al., 2020). The effectiveness of post-prandial fructose in conditioning sugar preferences in humans is uncertain because the few published human conditioning studies all used maltodextrins.

6. PLOS authors have the option to publish the peer review history of their article (what does this mean?). If published, this will include your full peer review and any attached files.

Reviewer #1: No

---

## [Author Response · Author response to Decision Letter 0]

17 Aug 2021

1. Your current title is: "Evidence that Human Oral Glucose Detection Involves A Sweet Taste Pathway and A Glucose Transporter Pathway." This title implies that you have identified two pathways for detecting glucose in the oral cavity. This is not the case. What you have established is that one can systematically alter glucose detection thresholds by modulating the activity of SGLT1. While I do not mean to diminish the importance of this finding, you currently do not know whether modulating the activity of SGLT1 alters downstream voltage-sensitive events in the T1r2-r3 pathway or whether it activates a T1r2-r3-independent pathway. Thus, I recommend that you scale back the scope of your title.

We respectfully disagree. We can take out T1R2/3 with lactisole and still have glucose transduction occur. We can also take out SGLT with phlorizin and still have sucralose be sweet. Therefore, the T1R2/3 and SGLT participate forin glucose transduction and sweet taste transduction with inhibition of the other and are, therefore, independent. They may converge at some point, but at least initially they are able to function are two independently of each other as taste systems. 

2. End of first paragraph of Introduction: You state: "Clearly sugars engage a second signaling pathway for calories that noncaloric sweeteners fail to engage post-prandially." I do not think that this statement follows logically from the five reasons you provide for why "...diet sodas have never captured a major share of the beverage market." Reviewer 1 has raised additional concerns about this statement.

We changed to: We hypothesize an additional explanation that sugars may engage a second oral signaling pathway for calories that noncaloric sweeteners fail to engage..

3. Line 1, second paragraph of Introduction: Change "mice" to "mouse"

OK

4. Line 4, second paragraph of Introduction: The following phrase seems awkward: "...the transportation of glucose". I think that it is more conventional to say "...the transport of glucose"

OK

5. Line 5, second paragraph of Introduction: Why don't you refer directly to the GLUTs? That would seem to be clearer than a vague reference to "...other transporters". 

We changed to…..Other transporters, such as GLUTs.

6. Lines 5-6, second paragraph of Introduction: I think that the following wording is accurate: "...the conversion of glucose into several ATP molecules..." Glucose is not actually converted into ATP molecules. The energy in the glucose molecule is used by glycolysis, krebs cycle and oxidative phosphorylation to add high-energy phosphate bonds to AMP and ADP.

Changed to "...the oxidation of glucose to produce several ATP molecules..."

7. Line 10, second paragraph of Introduction: I recommend changing "mouse tissue" to "mouse taste bud cells." 

OK

8. Lines 10-13, second paragraph of Introduction: I do not think that the prior literature has established that "taste bud cells in the mouth of mice are capable of identifying when a “sweetener” is metabolizable." Instead, I think this literature has established that murine taste bud cells may be able to detect elevations in salivary glucose via a T1r2+r3-independent mechanism.

We are asked to switch the verb capable to the verb able. Given our wording that the "system appears capable" we believe our wording is still appropriate.

9. In the last paragraph of the Introduction, you state: "Were such a signaling system to exist and be functional in the human mouth, it would have implications for oral signaling of metabolizable sugars and could help explain why humans overwhelmingly prefer sugared beverages to non-caloric beverages." I recommend that you dial-back this statement a bit for two reasons. First, because your paper addresses threshold (not suprathreshold) effects of SGLT1, you are creating false expectations in your readers. Second, the use of "metabolizable sugar" is a bit confusing. As noted by Reviewer 1, fructose may not be metabolized in taste cells, but it is certainly metabolized in the liver. Thus, fructose is a metabolizable sugar that does not activate SGLT1.

We changed sugars to glucose and emphasize preference here as an implication of our work rather than suprathreshold sweetness.

Changed to: Were such a signaling system to exist and be functional in the human mouth, it would have implications for oral signaling of the metabolizable sugar glucose and could help explain preference for sugared beverages over non-caloric sweetener beverages.

10. I was surprised that you did not incorporate the findings from a recent mouse paper [Yasumatsu K, Ohkuri T, Yoshida R, Iwata S, Margolskee RF, Ninomiya Y (2020) Sodium-glucose cotransporter 1 as a sugar taste sensor in mouse tongue. Acta Physiologica 230: e13560. doi: 10.1111/apha.13529], as it is highly relevant and supportive of your general hypothesis.

We cite this paper now. 

"This research was funded in part by a grant from the Suntory Global Innovation Center Limited

to PASB. The funder consulted on the general conception of the study and provided support in

the form of salaries for the authors (PASB, AT, LJF) and a scientist from Suntory (AI) helped

collect data under the supervision of the Monell Center. The specific roles of the authors are

articulated in the “Author Contributions” section. The funder did not play a role in the specific

study design, data analyses, decision to publish, or preparation of the manuscript."

"The funder did not play a role in the specific study design, data analyses, decision to publish, or preparation of the manuscript. "

The revised funding statement should read: 

This research was funded in part by a grant from the Suntory Global Innovation Center Limited to PASB. The funder consulted on the general conception of the study and provided support in the form of salaries for the authors (PASB, AT, LJF), and a scientist from Suntory (AI) helped collect data under the supervision of the Monell Center. The funder did not play a role in the specific study design, data analyses, decision to publish, or preparation of the manuscript.

Add to online-submission form

Additionally, because some of your funding information pertains to commercial funding, we ask you to provide an updated Competing Interests statement, declaring all sources of commercial funding.

In your Competing Interests statement, please confirm that your commercial funding does not alter your adherence to PLOS ONE Editorial policies and criteria by including the following statement: "This does not alter our adherence to PLOS ONE policies on sharing data and materials.” 

as detailed online in our guide for authors http://journals.plos.org/plosone/s/competing-interests. If this statement is not true and your adherence to PLOS policies on sharing data and materials is altered, please explain how.

 Please include the updated Competing Interests Statement and Funding Statement in your cover letter. We will change the online submission form on your behalf.

5. Review Comments to the Author

Reviewer #1: This revised manuscript is substantially enhanced by the addition of MDG and fructose experiments which support the involvement of SGLT1 sensing in oral glucose detection in humans. However, the characterization of this oral SGLT1-glucose sensing pathway as a "caloric" or " metabolic signaling pathway" is misleading. The SGLT1 pathway does not respond to fructose which is a "caloric" and "metabolizable" sugar. It is also unclear how much sucrose activates the SGLT1 taste pathway given that sucrose must be first hydrolyzed to glucose in the mouth to have this action. The authors should revise their description of the SGLT1 taste pathway. Also, for completeness, the authors should mention that galactose is a ligand for SGLT1 although, given the relatively low amount of free galactose in foods, this sugar may have minimal effects on the oral SGLT1 pathway.

We now state near the end of the discussion:

Logically, it is likely that other sugars besides glucose, such as galactose, can be transported by SGLT in taste tissue (Harada et al., 2012), and sugars that do not engage SGLT, such as fructose, may be transported via GLUT2 and GLUT8 in taste tissue (Sukumaran et al, 2016) and oxidized in taste tissue to produce ATP and close KATP channels (Berger et al,, 2020). 

The authors describe MDG as a ligand for SGLT1 but don't mention that it is also a ligand for the T1R2/T1R3 receptor and thus has a sweet taste to humans as it apparently has to mice as judged by their avidity for this glucose analog. The authors speculate that the SGLT1 taste pathway may contribute to the sweetness (or palatability) of sugars in humans and contribute to their preference for sugars over non-nutritive sweeteners. However, this does not appear to be the case in mice. SGLT1 KO and WT mice showed identical preferences for MDG over water in 3-min, 2-bottle tests (Sclafani, Koepsell and Ackroff, 2016). Furthermore, SGLT1 KO and WT mice showed similar preferences for a glucose + saccharin solution over a saccharin in a 1-h choice test. The KO mice consumed less glucose and MDG than WT mice which was attributed to their failure to normally absorb these sugars in the intestine. The authors should acknowledge that the oral glucose-sensing SGLT1 pathway does not appear to mediate oral sugar preference in mice although, as discussed next, it is essential for post-oral glucose preference conditioning. This does not preclude them from speculating about a role of SGLT1 taste pathway in sugar preference in humans.

We now acknowledge that SGLT1KO mice have sugar preference unaffected but still speculate about human oral influences on preference for sugars. We write: It is possible that oral SGLT contributes to glucose reward in humans either directly or indirectly via anticipatory metabolic regulatory reflexes, but not that oral SGLT in mice does not appear to contribute to preference (Sclafani et al., 2016).

The following statement in the Introduction should be supported with citations: "Clearly sugars engage a second signaling pathway for calories that noncaloric sweeteners fail to engage post-prandially." 

We now cite the paper showing that T1R KO mice have no responses to non-caloric sweeteners but continue to show responses to glucose and sucrose. Margolskee et al.

Presumably, the authors are referring to rodent and human studies showing that the post-oral actions of glucose and glucose-containing carbohydrates (sucrose, maltodextrin) condition preferences. However, describing this post-prandial pathway as "for calories" is questionable given the many rodent studies showing that glucose is much more effective than fructose in activating the post-prandial pathway. Of particular relevance here are studies showing that mice learn to prefer glucose but not isocaloric fructose over non-nutritive sweeteners (sucralose, Ace K), an outcome thought to be mediated by intestinal SGLT1 signaling (e.g., Sclafani & Ackroff, 2017; Tan et al., 2020). The effectiveness of post-prandial fructose in conditioning sugar preferences in humans is uncertain because the few published human conditioning studies all used maltodextrins.

In this context, we avoid discussing post-prandial feedback mechanisms in humans. 

---

## [Editor Report · Decision Letter 1]

20 Aug 2021

Evidence that Human Oral Glucose Detection Involves A Sweet Taste Pathway and A Glucose Transporter Pathway

PONE-D-21-16275R1

Dear Dr. Breslin,

We’re pleased to inform you that your manuscript has been judged scientifically suitable for publication and will be formally accepted for publication once it meets all outstanding technical requirements.

Kind regards,

John I. Glendinning, PhD

Academic Editor

PLOS ONE
---

## [Editor Report · Acceptance letter]

10 Sep 2021

PONE-D-21-16275R1 

Evidence that Human Oral Glucose Detection Involves A Sweet Taste Pathway and A Glucose Transporter Pathway 

Dear Dr. Breslin:

I'm pleased to inform you that your manuscript has been deemed suitable for publication in PLOS ONE. Congratulations! Your manuscript is now with our production department. 

Kind regards, 

on behalf of

Dr. John I. Glendinning 

Academic Editor

PLOS ONE